# Electro-Design of Bimetallic PdTe Electrocatalyst for Ethanol Oxidation: Combined Experimental Approach and Ab Initio Density Functional Theory (DFT)—Based Study

**DOI:** 10.3390/nano12203607

**Published:** 2022-10-14

**Authors:** Andile Mkhohlakali, Xolile Fuku, Min Ho Seo, Mmalewane Modibedi, Lindiwe Khotseng, Mkhulu Mathe

**Affiliations:** 1Analytical Chemistry Division, Mintek, 200 Malibongwe Drive, Randburg 2194, South Africa; 2Department of Chemistry, University of the Western Cape, Private Bag X17, Bellville, Cape Town 7535, South Africa; 3Institute of Nanotechnology and Water Sustainability, College of Science, Engineering and Technology, University of South Africa, Florida Science Campus, Roodepoort 1710, South Africa; 4Department of Nanotechnology Engineering, Pukyong National University, 45 Yongso-ro, Nam-gu, Busan 48547, Korea; 5Council for Scientific and Industrial Research (CSIR), Energy Center, Pretoria 0012, South Africa; 6Department of Chemistry, ICES, CSET, University of South Africa, Florida Science Campus, Roodepoort 1710, South Africa

**Keywords:** underpotential deposition, PdTe nanofilms, ethanol oxidation reaction, DFT calculation, oxygen binding energy

## Abstract

An alternative electrosynthesis of PdTe, using the electrochemical atomic layer deposition (E-ALD) method, is reported. The cyclic voltammetry technique was used to analyze Au substrate in copper (Cu^2+^), and a tellurous (Te^4+^) solution was used to identify UPDs and set the E-ALD cycle program. Results obtained using atomic force microscopy (AFM) and scanning electron microscopy (SEM) techniques reveal the nanometer-sized flat morphology of the systems, indicating the epitaxial characteristics of Pd and PdTe nanofilms. The effect of the Pd:Te ratio on the crystalline structure, electronic properties, and magnetic properties was investigated using a combination of density functional theory (DFT) and X-ray diffraction techniques. Te-containing electrocatalysts showed improved peak current response and negative onset potential toward ethanol oxidation (5 mA; −0.49 V) than Pd (2.0 mA; −0.3 V). Moreover, DFT ab initio calculation results obtained when the effect of Te content on oxygen adsorption was studied revealed that the d-band center shifted relative to the Fermi level: −1.83 eV, −1.98 eV, and −2.14 eV for Pd, Pd_3_Te, and Pd_3_Te_2_, respectively. The results signify the weakening of the CO-like species and the improvement in the PdTe catalytic activity. Thus, the electronic and geometric effects are the descriptors of Pd_3_Te_2_ activity. The results suggest that Pd_2_Te_2_ is a potential candidate electrocatalyst that can be used for the fabrication of ethanol fuel cells.

## 1. Introduction

Electrodeposition of thin films and nanostructured metal chalcogenides plays an important part in technological development. Nanostructured metal chalcogenides have been extensively studied, as they are characterized by fascinating properties that are different from the properties of the bulk counterparts. Transition metal chalcogenide-based tellurides have attracted immense interest in the field of catalysis, energy conversion, storage [1], and photo-electrocatalysis, as they exhibit high electrical conductivity and can be used in the field of molecular structure catalysis [2,3,4]. Among electrocatalysis reactions, ethanol (EtOH) oxidation reaction (EOR) has been a subject of interest, as a high theoretical density (8.01 kWhg^−1^) can be realized [5]. Among the direct alcohol fuel cells (DAFCs) known to date, direct ethanol fuel cells (DEFC) are widely studied, as they are less toxic, cost-effective, and present in abundance [3,6,7,8]. However, it is a challenge to break the C-C bond to realize the complete conversion of EtOH to CO_2_. The conversion process is accompanied by the release of 12e^−^ [6]. It has been reported that EOR intermediates such as CO can readily reduce the catalyst activity [3,7]. It has been previously reported that the addition of a second metal [7,8] such as Sn, Bi, Te, Pt, or Pd as promoter species improves the electrochemical performance of the system [9] toward EOR [8,9]. Researchers have tried to develop a rational design of Pd-based [2] electrocatalysts. The microwave polyol technique was used by Huang and co-workers to form PdTex/C for the application of ethanol oxidation reaction (EOR) in alkaline electrolytes [5,10]. Cai and co-workers also reported the synthesis of PdTe/C for EOR in alkaline media [3]. Under these conditions, Te improves the activity of Pd. A series of fabrication techniques such as chemical reduction and vapor phase deposition such as chemical vapor deposition (CVD), physical vapor (PVD), metal-organic chemical vapor deposition (MOVD), molecular beam epitaxy (MBE), microwave polyol, and hydrothermal method were reported. These methods focused on the process of electro-formation of transition metal chalcogenides such as HgCdTe, (CuInGaSe) and Pd-Te_x_/C [3]. However, the applications of these methods are limited, and this can be attributed to the epitaxial difficulties faced and the diffusion of the adlayer.

The electrochemical deposition method is a simple, versatile, and cost-effective method that can be used to form thin films. E-ALD has been utilized to fabricate metal films, superlattice thin films, and nanowires [11]. Recently, extensive research has been conducted on the process of the electro-formation of noble metal-based thin films by conducting surface-limited redox replacement reactions (SLRRRs). The fabricated systems can be used in the field of electrocatalysis. Mkwizu and co-workers used the E-ALD method to fabricate PtRu for methanol oxidation reaction and oxygen reduction reaction in acidic electrolytes [12]. Xaba et al. prepared PdBi and PdSn for alcohol oxidation reactions (AOR) in alkaline media [13]. The SLRR of Cu-_UPD_ (used for the fabrication of platinum group metal (PGM)-based electrode catalysts) was reported by Brankovic and co-workers [14]. Flowers and co-workers were the first to report the reductive Cd-_UPD_ and oxidative Te-_UPD_ characteristics that could be exploited to realize the electrodeposition of chalcogenides and other heavy metals [15,16]. A clear understanding of the correlation between electronic structures and alloys may provide insights into the behavior of electrocatalysts [17]. To the best of our knowledge, there are no reports on PGM-based chalcogenides, particularly on the synthesis of PdTe (for EOR in alkaline electrolytes) following the E-ALD method. The DFT and ab initio calculation methods have been used to verify the experimental results for TePd alloy toward EOR [18,19]. This method has rarely been used to study the electrocatalytic behavior of PdTe.

We used the DFT technique to determine the mechanism associated with EOR executed using PdTe electrodeposited following the E-ALD method. The theoretical study provides a better understanding of the effect of Te on bulk Pd that is used for EOR and the effect of surface structure on the electrocatalytic activity. Furthermore, the mechanism associated with PdTe-based EOR, oxygen binding energy, and data on the d-band center (influenced by the Pd:Te ratio (Pd, Pd_2_Te_2_, and Pd_3_Te_1_)) are reported. The theoretical and experimental results were compared, and the results agreed with each other. In addition, morphological characterization was realized using AFM and SEM techniques.

## 2. Materials and Methods

### 2.1. Reagents and Substrate Pretreatment

The precursor solutions for Te and TePd were prepared using deionized water (purity: >18.2 µScm^−1^). Methods previously reported by us were used for sample preparation [11]. (i) Cu solution (1 mM CuSO_4_ and 0.1 M H_2_SO_4_) at pH = 1.4, (ii) Pd solution (1 mM PdCl_2_ solution and 0.1 M H_2_SO_4_) at pH = 1.2, and (iii) Te solution (0.5 mM HTeO_2_^+^ and 0.1 M HClO_4_) at pH = 1.21 were prepared. Au was used as the substrate after soaking it in concentrated nitric acid and rinsing using water. Thereafter, the Au substrate was activated by cycling from 1.4 V to −0.2 V (30 cycles) in a solution of HClO_4_.

### 2.2. Electrochemical Characterization

The electrochemical characterization method was conducted to analyze the deposition potentials and electrocatalyst performance. AutoLab potentiostat (Metrohm-PGSTAT 302, Metrohm Autolab B.V. Kanaalweg 29-G 3526 KM Utrecht, The Netherlands) was used for analysis. The reference electrode was the saturated Ag|AgCl|3 M KCl, the working electrode was gold (Au) coated on glass (2 × 2 cm^2^), and the counter electrode was an Au wire embedded in PLEXIGLAS®. The performance of the electrode catalyst toward EtOH solution containing 0.5 M KOH and 0.1 M EtOH was determined using the CV technique (the voltage was cycled between −1.0 and +0.2 V). Prior to experimental analysis, the EtOH solution was de-aerated using N_2_ (g). CA and EIS techniques were used to examine the stability of the system and study the electron kinetics.

### 2.3. Physical Characterization

The X-ray diffraction (XRD) technique was used for analysis. The X-ray Rigaku Ultima IV X-Ray DI powder system was used to record the profiles. Cu Kα was used as the source to analyze the crystal structure and phase of the samples. The structural morphology of the samples was studied using a scanning electron microscope (SEM). The morphology and surface topography were examined using the atomic force microscopy (AFM) technique (Agilent Technologies AC Mode III instrument, Santa Clara, CA, USA), ambient tapping mode, using the RTESPW tip (Veeco Manufacturing, Inc., Plainview, NY, USA) model.

## 3. Results and Discussion

### 3.1. Cyclic Voltammetry of Au Coated Glass Recorded in HClO_4_ Solution Te ^4+^ and Cu^2+^ Ions

The CV technique was used to examine the UPD chemistry range of the metal. The E-ALD program used to deposit Te, Pd, and TePd on Au was studied, and the probable deposition mechanism was revealed. The CVs of the HTeO_2_^+^ and Cu^2+^ precursor solutions were recorded. Figure 1a shows the CVs of Au (in Te^4+^ solution), and the initial potential was 1.0 V. The final potential was varied during cycling (−0.2 V, −0.4 V, and −0.6 V). The reduction potential of the tellurous solution proceeds over multiple steps to produce telluride, a soluble (H_2_Te) species.

The Te reduction–oxidation reaction is expressed as follows Equation (1):(1)$$\begin{array}{c} HTeO_{2}^{+}+4e^{-}\to Te_{\left(0\right)}+2e^{-}\to H_{2}Te\;\\ or\\ \;HTeO_{2}^{+}+6e^{-}\to H_{2}Te \end{array}$$

Figure 1a presents four progressive reductive characteristics observed along the negative scan direction. The peaks at +380 mV and −400 mV can be attributed to Te-_UPD_, and bulk Te_(0)_ is believed to deposit at approximately −500 mV. The peak at −580 mV is attributed to the onset of hydrogen evolution (H_2_-evol), and intense H_2_-evol is observed at −770 mV. Bulk Te-strip contributes to the oxidative current peaks appearing during cathodic scans at +450 mV and +600 mV. These peaks can be attributed to the oxidation of Te-_UPD1_ and Te-_UPD2_, respectively. The redox characteristic corroborates tellurium deposition in acidic electrolytes, as illustrated by the reduction reaction mechanism reported in the literature [20]. Conclusively, the preference for using Te (OPD = −600 mV) can be attributed to the slow kinetics of Te-_UPD_.

Figure 1b presents data on the Au substrate in solution of 1 mM CuSO_4_ + 0.1 M HClO_4_ (cycled from +0.6 V to −0.2 V). The presence of two prominent peaks (at 0.180 V and −0.099 V) corresponding to the system under study has been reported in the literature [B]. The peak corresponding to Cu-_UPD_ is more intense than the peak corresponding to Cu(_0_) counterpart, and this can be attributed to the strong metal substrate (M–S) interactions (between Cu-_UPD_ and the metal substrate) [21]. This indicates that the strength of the bond formed between Cu-_UPD_ and the Au substrate is stronger than the strength of the metal–metal (M–M) bond. The results were recorded around the overpotential deposition (OPD) potential [22]. The standard reduction potential and Cu underpotential deposition are well presented in literature reports [11]. The standard reduction potential corresponding to Cu has been presented in the Appendix A. The shapes of the CVs and the peaks observed were almost the same as those reported previously [13,18,19,20,21,22,23,24,25,26].

### 3.2. Electrodeposition of Monometallic (Te, Pd) and Bimetallic (PdTe, in Various Te Ratios) Nanofilms 

Previously reported electrodeposition methods were followed for the electrodeposition of thin films [11,13,27]. The process of depositing Te on Au has been reported in the literature [11], and the schematic representation of the deposition process is presented in Figure 2a,b. The TePd E-ALD formation cycle proceeds over several steps: (i) Cu = open-circuit potential (OCP), 10 s; (ii) Cu-UPD = +180 mV, 10 s; (iii) Pd = OCP, 10 s, (iv) the BE = OCP, (v) Te = OCP, 15 s, (vi) held Step 4 quiescent to permit Te bulk deposition of at −500 mV, and (vii) BE = −600 mV (allows the reduction of bulk Te keeping the atomic Te layer intact). The last step is associated with oxidative Te underpotential deposition (oxTe-_UPD_). This process was repeated over Te cycles to generate different Pd:Te ratios (Pd_3_Te_1_, 10 Te cycles; Pd_3_Te_2_, 15 Te cycles).

Figure 2 presents the method of formation of Te and PdTe. It can be observed that when the Te solution is flushed at OCP, no current spike is generated. When Te is deposited at OPD (−500 mV), a prominent reductive current spike is generated, which indicates the occurrence of the process of bulk Te deposition, as denoted in Figure 2a. When the potential for Be flow is set at −600 mV, an oxidative current spike is generated. It was believed that small amounts of bulk Te are stripped off under these conditions, and Te adatoms are left behind. These steps corroborate the results obtained by analyzing Equation (1), and this explains the redox couple observed for oxidative Te-_UPD_ and the process of re-reduction of bulk Te(0). The formation of bulk and soluble tellurite (H_2_Te) species is presented in Figure 2b. Figure 2 also presents the SLRRR process occurring at the beginning of the cycles. Similar results have been previously reported by our group [11,28]. A negative (reductive) current shift was observed during the Cu-_UPD_ step at 180 mV. This reflects the deposition of the Cu-atomic layer. The SLRRR process corresponding to the interchange of Cu-_UPD for_ Pd at the open-circuit potential (OCP) has been reported previously by us [11]. The results are consistent with previously reported results [11,28,29]. The E-ALD model was developed following the methods reported in the literature [11]. A distinct change in current between −1000 and 0.1 µA is observed when BE flow is observed at −500 mV. The spike in the peak is believed to correspond to oxidative current response. This indicates the presence of oxidative Te-_UPD_ adatoms on the surface of Pd. The OCP values increase with an increase in the number of cycles associated with Te deposition on Pd. This is revealed by the Labview sequencer 4 software. This potentially indicates that the process of Te deposition results in varying degrees of surface coverage. Deposition occurs on Pd and Te-modified Pd surfaces.

### 3.3. Morphological Characterization

#### SEM and AFM Analysis

Figure 3a–c presents the SEM micrographs of the samples: Pd, Pd_3_Te_1_, and Pd_2_Te_2_ nanofilms. Pd consists of flat conformal quasi-spherical particles deposited across the Au substrate. Analysis of the Au nanostructure indicates the presence of epitaxial characteristics. The results agree well with the results reported in the literature [3]. It was observed that Pd_3_Te_1_ consists of two phases (white and gray particles). The generation of the two phases could be attributed to the characteristics of the Pd and Te nanostructures. The white particles show the characteristics of typical Te-dendritic burls. These particles are cauliflower-shaped, and similar results were reported by Ma et al. [30]. Interestingly, Pd_3_Te_2_ deposits present flower-shaped (fern fronds) profiles that are present in abundance throughout the surface. The microstructures observed for these samples were similar to the Te-feather-like microstructures reported previously [31]. The dendrites grow from the core (nuclei) in different directions. These consist of distinct branches of different sizes. The characteristics agree well with the characteristics of the deposited dendritic Te particles reported in the literature [32]. Notably, the branches increase in size as the number of Te cycles increases (15 cycles) (Figure 3c). The increase in the sizes of the branches resulted in the formation of a large number of irregular trunks that overlap with each other at angles in the range of 45–60°. The ends of the sub-branches appear cauliflower-like. These contain distinct layers, the formation of which can be attributed to the layer-by-layer Te-deposition method. Conclusively, Te plays a crucial role in alternating the structural morphology of the Pd surface. This indicates that the surface characteristics depend on the number of ox-Te-_UPD_ cycles. The characteristics are influenced by the surface-directing diffusion process. Furthermore, the incorporation of Te into Pd was confirmed by the morphology distortion process that resulted in the generation of dendritic structures, the formation of rugged surfaces, and the generation of grain boundaries. 

Figure 3d–f reveals that the Pd samples present flat and uniform surfaces. The surface characteristics can be potentially attributed to the layer-by-layer epitaxial 2D growth process promoted by Pd SLRR. Pd_3_Te_1_ grains sit on top of each other, and this indicates early nucleation and 3D growth. These processes result in the formation of an island. Te-containing Pd presents a rough surface, and coalesced particles attributable to the nanostructures of neighboring Te and Pd units were observed. Grain boundaries were observed for the case of Pd_3_Te_2_, and the formation of grain boundaries can be attributed to the deposition of Te at distinct active sites (Pd remnant site and Te-covered Pd sites). The effect of the Te ratio on the morphology was studied, and it was observed that the roughness increases with an increase in the ratio. This can be attributed to the deposition of the overpotential deposition (OPD) steps associated with Te. The nanofilms differ in roughness. The average surface roughness (Sa) is calculated for Pd (Sa = 21.59 nm), Pd_3_Te_1_ (Sa = 32.50 nm), and Pd_3_Te_2_ (Sa = 45.5 nm). The results reflect the SEM results presented in Figure 3a–c.

### 3.4. Electrochemical Performance of Pd-Based Electrode Catalysts and Their Kinetics during EOR

The Pd-based electrode catalysts examined were Pd, Pd_3_Te_2_, and Pd_3_Te_1_ (Figure 4a). The cycling experiments were conducted between −1.0 and +0.2 V. Analysis of the CVs presents the characteristic features of Pd in a solution of KOH. The results agree well with the results reported in the literature [28]. During the process of cathodic scan, the CVs exhibit intense and broad peaks in the regions spanning from −0.2 to −0.18 V and −0.83 V to −0.99 V. The origin of the peaks was attributed to the reduction and oxidation of Pd-O and hydrogen evolution (adsorption-desorption), respectively. Voltammograms recorded under conditions of positive (anodic) scan depict the onset of oxidation. The potential varied from −0.1 V to +0.05 V, and this was ascribed to the process of Pd-O/Pd-OH monolayer formation [33]. Pd_3_Te_1_ generates a high current response, whereas Pd_3_Te_2_ exhibits a broad current response under conditions of Pd-O reduction, and oxidation peaks are generated. These can be potentially attributed to the strong electronic interactions between Te and Pd. The increased Pd-O current and the negative shift observed for the unmodified Pd counterpart reveal that significantly high degrees of interactions are present between the Te electrons and Pd. The addition of Te in the crystal structure of Pd can also explain the observations. Moreover, the impact of introducing Te on the surface of Pd increases the amount of the oxide layer, and this can be attributed to the fact that Te is an oxygen species [3]. 

Two different Pd:Te ratios were investigated towards EOR as illustrated in Figure 4b. It was observed that the introduction of Te on Pd electrode catalysts reduced the onset potential and increased the oxidation current during EOR as observed in the forward peak current (i_f_). This indicated a reduction in the activation energy and the rapid shuttling of electrons. The processes were induced by the modifications on the Pd surface with Te adatoms. The results can also be explained by the Te oxyphilic features. These features result in the generation of M(OH)_x_ species at lower potential regions [34]. Te, as an oxophilic metal, may help to further oxidize the CO (EOR intermediates) species. Oxidation of this species can be promoted by the synergistic effect between Te and Pd. The generation of the synergistic effect facilitates the bifunctional mechanism and electronic effect, as described in previously reported papers [6]. The sharp reverse peak (backward peak current (i_b_)) is believed to reflect the high oxygen binding energy on the surface of Pd. The center of the d-band of Pd shifts under these conditions, and eventually, the species is oxidized further, resulting in the accumulation of EOR byproducts. The results are in strong agreement with DFT calculation results. A summary of the electrochemical activities for EOR is presented in Table 1.

The stability and the electron mobility of the best-performing catalyst (Pd_3_Te_2_ in 0.5 M KOH) were investigated, as displayed in (Figure 5a). It is observed that the palladium reduction (PdO_r_) peak increases with an increase in the scan rate under conditions of the cathodic scan. The results reflect the stability and the occurrence of the diffusion-controlled reactions on the electrocatalyst. The durability of Pd_3_Te_2_ was further tested for EOR (0.5 M KOH and 0.1 M EtOH solution), and the results were compared with the results obtained for the Pd counterparts (Figure 5b). A rapid decay in current at the beginning (a few seconds) was observed for the electrocatalysts. This can be attributed to the accumulation of intermediate species (produced during EOR) on the active sites of the surface of the electrocatalyst. The polarization current decay reached a steady state, and at this point, a 2-fold increase in current was observed for Pd_2_Te_2_. The magnitude varied in the order Pd_2_Te_2_ > Pd > Pd_3_Te_1_. This suggests that Pd_3_Te_2_ is highly tolerant toward the CO_ads_ intermediate. The high tolerance can be potentially attributed to the incorporation of the Te (oxygenated) species. During the last few minutes, Pd_3_T_1_ overlaps with the Pd catalyst, resulting in current decay. The observation may be better described by the geometric effect or the low structural stability and inhomogeneous (agglomeration) surface morphology of the samples (AFM; Figure 3f). The results agree well with the results obtained using the DFT technique. The results explain the effect of the rugged surface and the oxygen binding energy recorded on the Pd surface. The results are in strong agreement with the results obtained by analyzing the CVs (Figure 4a,b).

### 3.5. Electrochemical Impedance Spectroscopy (Pd vs. TePd)

The best-performing catalyst (Pd_2_Te_2_) was placed under scrutiny to examine the interfacial electron transfer process and compare the results with the results obtained by studying the unmodified Pd sample. Figure 6a shows the EIS plots (illustrated using a Nyquist plot). It can be observed that the electrode catalysts were characterized by a distinct size of the impedance arc (DIA). It has been previously reported that the size of DIA predicts the charge transfer resistance [35,36]. The Randles–Sevcik equivalent circuit (Figure 6a: insert) has been used to simulate the EIS elements (R_s_(R_ct_CPE)) from the semicircles to obtain the values, and the associated errors obtained were less than 4% [37]. As displayed in Figure 6, the semicircle corresponding to the Pd electrode catalyst is bigger than that recorded for the PdTe counterparts. The size of the semicircles recorded for PdTe indicates a low charge transfer resistance and fast electron transfer process [38]. These results are affected by the modification of the Pd surface (by Te oxygen species). The liberation of Pd from CO (EOR byproduct) promotes the EtOH adsorption/dehydrogenation process, as explained in previously reported papers [11,25] and Appendix A. A bifunctional mechanism is thought to be associated with the process. This implies that Te oxygenated (Te promoter) species promote the process and help improve the electric conductivity of the bimetallic PdTe electrode catalyst. A rapid electron kinetic transfer is observed for this system, and the process is faster than the process observed for Pd [11]. These findings corroborate the results obtained using the CV and DFT techniques. The results revealed that the Pd-O bond strength could be improved, indicating the high electrocatalytic activity of the system during EOR.

Figure 6b illustrates the Bode plot recorded with Pd- and PdTe-based electrode catalysts. The plots were recorded in EtOH in an alkaline media. The PdTe-based electrocatalysts were characterized by a high maximum phase angle (-θ_max_) and metallic character (72° and 84.7°, close to 90°). The capacitance recorded was comparable to the total capacitances. The impact of Te on the surface of Pd was studied, and it was observed that Te modification helped increase the phase angle from −72° to 84.7°. This indicated an increased conductivity, a rapid electron transfer process, and a high electrocatalytic activity [39]. In addition, the electron shuttling process is promoted by the incorporation of Te, as revealed by the CVs presented in Figure 5b. Te insertion can potentially increase the concentration of the hydroxyl (^−^OH) anions on the Pd surface. This result may also indicate the strong oxygen binding property (unlike the case of Pd) of PdTe. Similar results were obtained using the DFT technique (Figure 6 and Figure 7). This may promote further attack and result in the stripping off of the intermediate poisonous carbonyl (CO) species [32]. This is also reflected by the bifunctional mechanism and the transformation of PdTe. The resistive electron movement observed at low-frequency values changes to capacitive behavior at high-frequency values. The results reflect the results obtained from the Nyquist plots (Figure 6a) and CVs (Figure 4b). These plots reveal a decrease in the charge transfer resistance (Rct). A high peak current that confirms electron mobility is observed in this case. The results were compared with the results obtained by studying Pd. 

### 3.6. Computational Details

The Vienna Ab initio Simulation Package (VASP) [26], based on the DFT technique [40,41], was used to simulate the ethanol oxidation activities of the PdTe alloy catalyst. The Kohn–Sham wavefunctions were solved to arrive at the results. The projector augmented wave (PAW) pseudopotential was used to replace the interaction between the core electrons [39,42]. The exchange-correlation energies of the electrons were described using Perdew, Burke, and Ernzerhof (PBE) [43]. The energies were revised using the Perdew–Burke–Ernzerhof functionals (RPBE) [44] under conditions of generalized gradient approximation (GGA) [45,46]. Pd was characterized by a traditional fcc structure. The system was designed, and the Te of the second (2nd) metal was replaced based on the ratio of Pd and Te. Five bulk structures were generated by substituting Te atoms at the edge and face sites (based on the Pd-to-Te ratio) to determine the configurations of the systems. Before designing the surface of Pd, Pd_x_Te_y_, each sample was fully relaxed in the bulk structure. The gamma point mesh was used (13 × 13 × 13; k-points), and the kinetic cutoff energy was set at 520 eV. The fully optimized bulk structures were cleaved into (111), (110), and (100) surfaces. The *z*-axis vacuum length was set at 20 Å to avoid interactions between the top and bottom surfaces. The bottom three layers were fixed to the surface, and the designed models were simulated using the gamma point (6 × 6 × 1) k-points. After obtaining the most stable surface structure, the oxygen atoms were placed on the surface of the possible configurations. Binding energies (ΔE_BE_) of oxygen were calculated on the Pd, Pd_x_Te_y_ (111) surfaces. The DOS values (Figure 7) were used to execute the tetrahedron method with Blöchl corrections [42] to better illustrate the structural interactions.

#### The First-Principle Studies on Pd and PdTe Surfaces Used for Ethanol Oxidation

Based on the ab initio study, the electronic structure of Pd and Pd_x_Te_y_ were calculated to reveal the relationship between the experimentally measured specific activity, the oxygen binding energies, and the positions of the d-band centers (Figure 8). 

Figure 8 presents the density of state profiles and the optimized Pd (111), Pd_2_Te_2_ (111), and Pd_3_Te_1_ (111) structures. The results were obtained using the ab initio DFT technique. The d-band center of Pd was calculated from the spectra, and the result indicated the surface reactivity for ethanol oxidation. The electron in the d-band corresponding to Pd in Pd_2_Te_2_ (111) and Pd_3_Te_1_ (111) was at a higher level than the electron in Pd (111). This could be attributed to electronegativity differences that could result in electron-charge transfer from the less electronegative Te (2.1) to the more electronegative Pd (2.2). These results are comparable to those obtained using the EIS and CV techniques. The position of the d-band center corresponding to the alloy surfaces differs significantly from the position of the d-band center of equivalent pure Pd (111). This can be attributed to d-band hybridization in the presence of Te atoms. The position of the d-band center of Pd (111) was determined at −1.83 eV (with respect to the Fermi level). The results agree well with the previously reported results [47,48,49]. The theoretical d-band center of homogeneous Pd_2_Te_2_ (111) and Pd_3_Te_1_(111) were found to be −2.14 and −1.97 eV, respectively (Figure 7a,b). These peaks shifted negatively with respect to the peak positions corresponding to Pd (111). Bader charge analyses [50] were conducted, and DOS corresponding to Pd_3_Te_2_ (111) and Pd_3_Te_1_ (111) were determined. The Bader charge density is calculated using Equation (2) as follows:(2)$$\begin{array}{c} \rho =\rho_{Pd_{x}Te_{y}}-\rho_{Pd_{x}}-\rho_{Te_{y}}, \end{array}$$

In order to verify the experimental results, the five bulk structures of PdTe were designed from the conventional Pd fcc structure. The atoms were substituted at the face or edge sites. Three models having the lowest DFT internal energy were selected (Figure 9a–d). The theoretically calculated lattice parameters for *x*-, *y*-, and *z*-axes in the ground state of each Pd alloy were 3.940 Å for Pd bulk, 3.579, 4.422, and 5.989 Å for Pd_2_Te_2_, and 4.092 Å for Pd_3_Te_1_, respectively. The lattice parameters corresponding to Pd_2_Te_2_ exhibit reduction along x, while elongation along the *y*- and *c*-axes can be attributed to the process of replacement of the face site by the Te atoms. Figure 9a presents the theoretically calculated XRD patterns of Pd, Pd_2_Te_2_, and Pd_3_Te_1_. The results agree well with the experimentally obtained results for the synthesized PdTe alloys, expanding lattice parameters with increasing Te contents in the bulk lattice, as shown in Appendix A.

As shown in Appendix A, the experimental XRD patterns is recorded for monometallic (Pd) and bimetallic (PdTe). The samples prepared by varying the Pd:Te ratio were analyzed. Four primary (Bragg’s diffraction) peaks were observed in the profiles at 38.3° (111), 51° (200), 73° (220), and 80° (311) (this corresponds to the face-centered cubic (fcc) characteristic of metallic Pd). The results agree well with previously reported results [28,47]. Peaks corresponding to bimetallic nanofilms shifted to lower 2-theta angles ((111) and (220)) (with respect to the Pd counterpart). This change can be attributed to the insertion of Te into the Pd crystal lattice and the strong interactions between the Pd and Te adatoms. These features indicate the characteristic formation process of an alloy, and the results agree well with the results obtained using the DFT studies. The strong Pd/Au (111) peak suggested that the deposits preferred the (111) orientation. The low-intensity peaks corresponding to the PdTe catalyst appearing at the low (25°, 20°, 16°) 2θ regions can be attributed to the presence of elemental Te. The results reflect previously reported results [31]. These findings were consistent, as displayed in Appendix A and as reported by this group in the literature [4]. These peaks are parallel to (111), indicating the existence of an epitaxial relationship between the catalyst and the substrate. 

The slab models were designed based on Pd, Pd_2_Te_2_, and Pd_3_Te_1_ (Figure 10b–d). To identify the most stable surface of the PdTe alloy, the surface structures were investigated by calculating the surface free energy. The (111), (110), and (100) surface models were fabricated, as shown in Figure 9, and the surface energies were obtained using Equation (3) (0.332, 0.405, and 0.533 $\mathrm{eV}\cdot {Å}^{-2}$ for Pd; 0.178, 0.209, and 0.254 $\mathrm{eV}\cdot {Å}^{-2}$ for Pd_2_Te_2_; 0.250, 0.303, and 0.422 $\mathrm{eV}\cdot {Å}^{-2}$ for Pd_3_Te_1_; Appendix A).
(3)$$\begin{array}{c} \sigma =\frac{1}{A}\left[E_{slab}-nE_{bulk}\right] \end{array}$$

Here, σ is the surface energy; A is the surface area exposed to vacuum on both sides of the slab. E_slab_ and E_bulk_ represent the total energies corresponding to the surface and bulk models of Pd, Pd_2_Te_2_, and Pd_3_Te_1,_ and n is the number of the unit cells in the perfect bulk crystal that matches the total number of atoms in the slab model. For all Pd_x_Te_y_ systems, the (111) slab models were thermodynamically stable. 

The charge density plots corresponding to PdTe (111) and Pd_3_Te_1_ (111) along the *z*-axes are presented in Figure 10c–f. The accumulation of the negative charge is represented by yellow, and the depletion of the negative charge is denoted in sky blue. The surface charge density corresponding to the two slab models was calculated by analyzing the charge density on the surface. Negative charge accumulated on Pd (−1.14 e), which is the catalytic active site as compared to less amount of negative charge accumulated at the depleted Te site (+0.12 e) on the PdTe surface. EOR in alkaline media has been explained by various research groups [11,51,52,53]. The process can be explained by the following half-cell reaction
(4)$$\begin{array}{c} C_{2}H_{5}OH+3H_{2}O\to 2CO_{2}+12H^{+}+12e^{-} \end{array}$$

C-C bond breaking may not be observed for EOR byproducts present at the Pd interface. This can potentially result in the generation of acetyl (CH_3_CO_ads_ a) products and not CO_2_. The mechanism that can best explain the adsorption and generation of species has been presented. The extent of CH_3_CO_ads_ and OH_ads_ coverage realized [54,55] influences the mechanism presented in the Appendix A and literature reports [7,54,55,56]. The rate-determining step was identified by analyzing Appendix A (Appendix A) [7,54]. The OH_ads_ units adsorbed on the electrode play an important role during EOR. The adsorption of the byproducts can be prevented in the presence of the adsorbed hydroxyl (-OH) species, and free active sites are regenerated on the catalyst under these conditions. 

Furthermore, it is important to study the oxygen adsorption energy to understand the catalytic activity for ethanol. The DOS of electrode catalysts at the Fermi level plays a crucial role in the chemisorption of the adsorbates. It significantly influences the performance of the catalyst. Therefore, the binding energy of oxygen was calculated based on the surface (111) structure of Pd, Pd_2_Te_2_, and Pd_3_Te_1_. The oxygen adsorption sites were investigated for possible locations depending on the top, hcp, and fcc structures, as shown in Figure 10. The three structures are presented in Figure 10b–d, and the energies were obtained using Equation (5) as follows: (5)$$\Delta E_{O*}=E_{Pd_{x}Te_{y}\left(111\right)-O*}-E_{Pd_{x}Te_{y}\left(111\right)}-E_{O}$$
where E_PdxTey(111)-O*_ denotes the oxygen adsorbed on the catalyst surface, E_PdxTey_ is the internal energy of the (111) slab models of Pd and Pd_x_Te_y_, and E_O_ is the DFT-based energy of the isolated single oxygen atom.

Analysis of the oxygen binding energies revealed that the higher the d-band center positions, the greater the bond energy between Pd and the oxygen species (e.g., OH^−^). Comparisons were made with respect to the pure Pd surface. The strong bond formed by oxygen can potentially result in the induction of a catalytic oxidation reaction. We observed that the d-band shifted downward, indicating the formation of weak bonds between oxygen species and Pd surface. Low activity is observed for the Pd_3_Te_1_ catalyst. This result shows the discrepancy with the experimentally obtained results. The overall activity of the catalyst is a cumulative effect of the geometric and electronic effects of the catalyst [57,58,59]. When the Te content with bcc structure increased, the lattice of the Pd_2_Te_2_ system was significantly distorted, and elongation along the y- and c-axes was observed (Figure 10c). Elongation results in an increase in the free surface energy of the rugged Pd_3_Te_2_ (111) system (Figure 10c). 

The geometric effect results in the generation of high binding energy for oxygen (Figure 8). This phenomenon represents that OH^−^ ions can be readily adsorbed onto Pd_3_Te_2_, and this was not the case for Pd. The OH_ads_ units adsorbed on the electrode will oxidize the ethanol byproducts such as CH_3_CO_ads_ and CH_3_COOH by interacting with the surface OH^−^ ions [55]. For the case of the Pd_3_Te_1_ catalysts, it was observed that the farther the d−band center from the Fermi level, the weaker the interaction of the system with the adsorbents (e.g., oxygen). This indicated that the catalytic activity of the system was lower than the catalytic activity of Pd. High binding energy was recorded for Pd_3_Te_2_, and this could be attributed to the geometric effect. However, a low-lying d-band position was recorded for this system. An increase in the Te content indicates a geometric effect. Under these conditions, the geometric effect is more dominant than the effect attributable to electronic structural changes. This reveals the presence of rugged surfaces that can potentially prevent the formation of EOR byproducts and reveal the active sites of the catalysts. These results are consistent with the experimental results that reveal that the catalytic activity increases during ethanol oxidation reaction (Appendix A). 

## 4. Conclusions

Electrodeposition of monometallic Pd and bimetallic PdTe (by varying the Pd-to-Te ratio) was successfully realized following the E-ALD method. The E-ALD cycles proceeded through multiple steps of SLRRR. Cu-_UPD_ was used for the deposition of Te in the presence of oxidative Te-UPD. The introduction of the Te adlayer on Pd improved the electro-oxidation property for EOR. This marks the development of the first PGM-chalcogenide (PdTe)-based electrocatalyst following the process of E-ALD. The bimetallic PdTe system exhibited improved catalytic activity and electron transfer kinetics (compared to Pd). The results were arrived at using the CV and EIS techniques. Analysis of the XRD patterns confirmed the formation of the PdTe alloy. The negative 2θ shift indicated the interaction between Te and the Pd crystal lattice. Te plays a crucial role in fine-tuning the structural morphology and the electrochemical performance of the Pd-based electrode catalyst. The insertion of Te was confirmed by studying the morphology distortion of the system. Dendritic structures and surface grain boundaries were formed, and these were obtained using the SEM and AFM techniques, respectively. The EOR mechanisms and electrocatalytic activity of Pd (111) and PdTe (111)-based catalysts were successfully investigated using the ab initio and DFT techniques. Strong oxygen bond energy was recorded following the incorporation of Te into Pd systems. This indicates the presence of weak CO-like species and improved activity. The DFT results corroborate the theoretically calculated results. A combination of geometric and electronic effects could effectively describe the effect of the Te content on the performance of the PdTe alloy. The electronic structural effect was found to be the dominant effect. The E-ALD method can be controlled to deposit PGM-based systems doped with chalcogenides for the fabrication of ethanol fuel cells.

## Figures and Tables

**Figure 1 nanomaterials-12-03607-f001:**
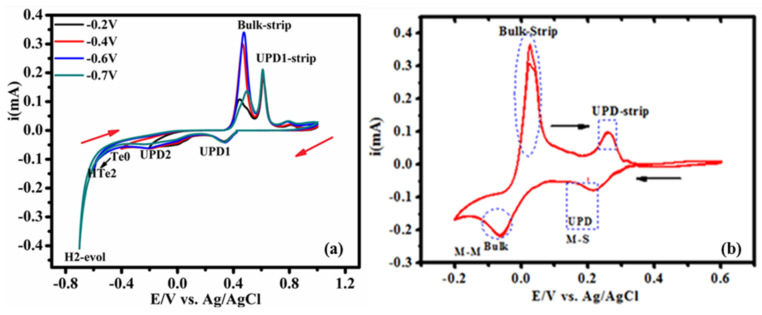
CV of Au recorded in aqueous solutions containing (**a**) 0.5 mM HTeO^+^_2_ + 0.1 M HClO_4_ and the profiles recorded in (**b**) 1 mM CuSO_4_ + 0.1 M HClO_4_.

**Figure 2 nanomaterials-12-03607-f002:**
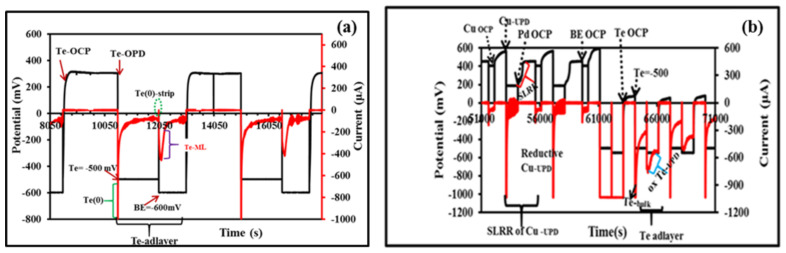
Potential–current–time plot generated for Te (**a**) and PdTe (**b**) over two E-ALD cycles.

**Figure 3 nanomaterials-12-03607-f003:**
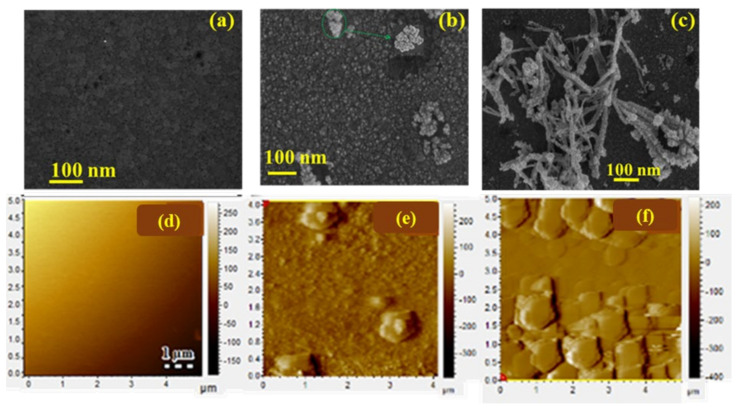
(**a**–**c**) SEM and (**d**–**f**) 2D AFM micrographs recorded for Pd (**a**,**d**), Pd_3_Te_1_ (**b**,**e**), and Pd_2_Te_2_ (**e**,**f**) nanofilms.

**Figure 4 nanomaterials-12-03607-f004:**
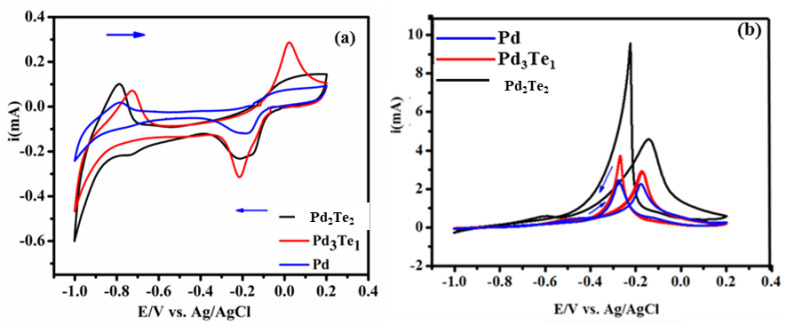
(**a**) CVs recorded for the Pd-based electrode catalysts in a solution of (**a**) 0.5 M KOH and (**b**) 0.5 M KOH + 0.1 M EtOH (scan rate: 30 mVs^−1^).

**Figure 5 nanomaterials-12-03607-f005:**
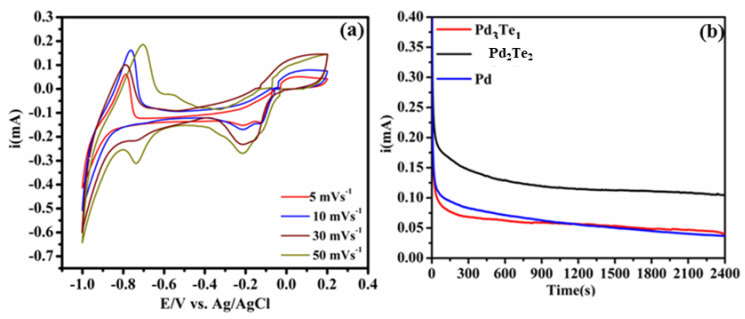
(**a**) CVs recorded for Pd_2_Te_2_ in 0.5 M KOH under conditions of varying scan rates and (**b**) CA of the Pd-based electrode catalysts recorded in a solution containing 0.1 M EtOH and 0.5 M KOH at the potential of −0.2 V (time: 3600 s).

**Figure 6 nanomaterials-12-03607-f006:**
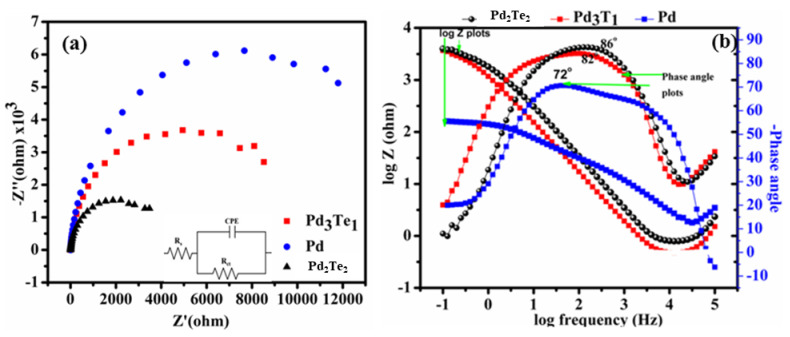
Nyquist plots generated for electrode catalysts in a solution of EtOH at −0.2 V and 1 × 10^4^–0.1 Hz. The insert presents the equivalent circuit used to fit the impedance curve.

**Figure 7 nanomaterials-12-03607-f007:**
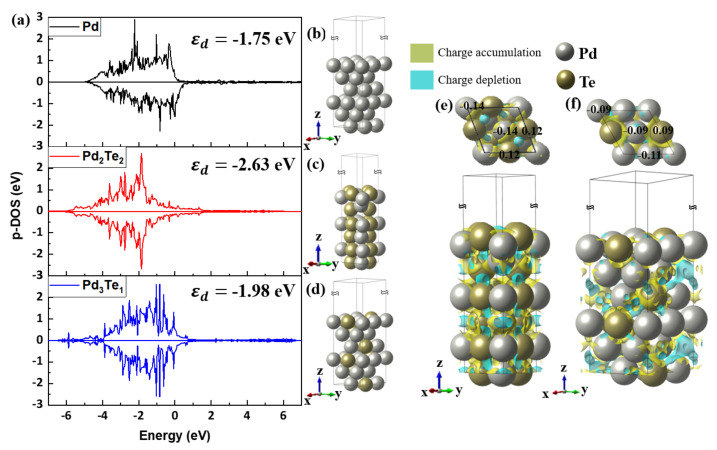
(**a**) Projected d-DOS spectral profiles recorded for the Pd atom. Data for the surfaces of (**b**) Pd (111), (**c**) Pd_2_Te_2_ (111), and (**d**) Pd_3_Te_1_ (111). Bader charge analysis and charge difference recorded for (**e**) Pd_2_Te_2_ and (**f**) Pd_3_Te_1_ slab models. The yellow and blue colors denote negative charge accumulation and depletion, respectively.

**Figure 8 nanomaterials-12-03607-f008:**
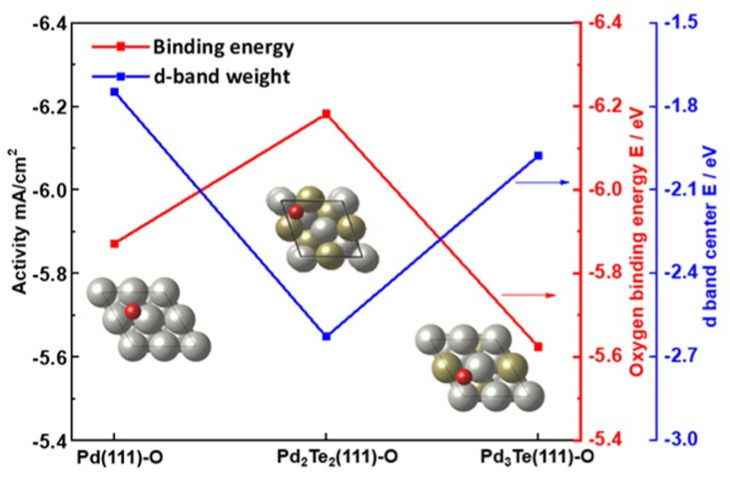
Theoretically specific activity recorded for ethanol oxidation and the oxygen binding energies and d-band center values recorded for the cases of Pd (111), Pd_2_Te_2_ (111), and Pd_3_Te (111).

**Figure 9 nanomaterials-12-03607-f009:**
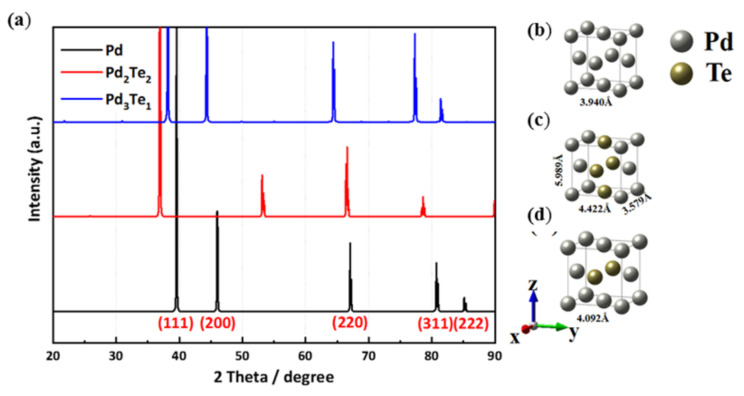
(**a**) Theoretically calculated X-ray diffraction patterns corresponding to (**b**) Pd, (**c**) Pd_2_Te_2_, and (**d**) Pd_3_Te_1_ (bulk structures).

**Figure 10 nanomaterials-12-03607-f010:**
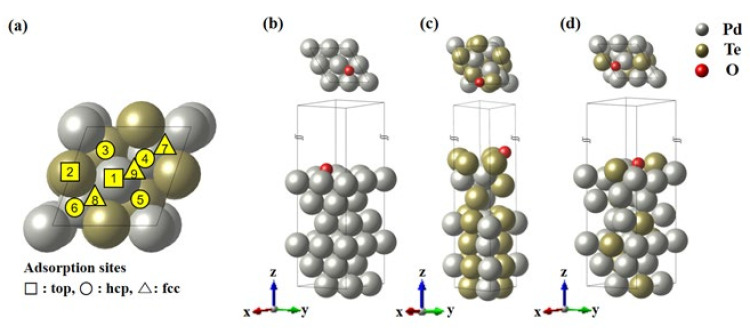
(**a**) Schematic representation of the oxygen adsorption sites on the Pd_x_Te_y_ (111) slab models. (**b**–**d**) Defined slab models containing oxygen at the (**b**) Pd (111), (**c**) Pd_2_Te_2_ (111), and (**d**) Pd_3_Te_1_ (111) fcc position.

**Table 1 nanomaterials-12-03607-t001:** Performance parameters of Pd-based electrode catalysts.

Catalyst (mV)	i_f_ (mA)	i_b_ (mA)	i_f_/i_b_ (mA)	Onset (E/V)
Pd	2.0	2.4	0.9	−4
Pd_3_T_1_	3.2	3.75	0.8	−0.42
Pd_2_T_2_	5.0	9.5	0.52	−0.48

## Data Availability

The data presented in this manuscript will be available upon request from the corresponding author.

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
