# Peer review of "Electro-Design of Bimetallic PdTe Electrocatalyst for Ethanol Oxidation: Combined Experimental Approach and Ab Initio Density Functional Theory (DFT)—Based Study"

_nanomaterials, 2022, doi:10.3390/nano12203607_

Round 1

Reviewer 1 Report

Review attached

Author Response

I appreciate your kind criticisms. The comments are addressed and discussed bellow;

Andile Mkhohlakali 1,5,* is inserted as suggested

Abstract

  1. ‘’Au substrate’’ is inserted and highlighted
  2. Pd is included

Introduction

All the abbreviation are defined first and are highlighted.

Materials and method

  1. (i) and (ii) are included before (iii) in section 2.1
  2. Au-coated glass was changed to Gold (Au) coated on glass and the dimension (2x2 cm2) is inserted.
  3. The ‘system is deleted in line 84. In addition, the sentence in section 2.2 is re-arranged according, and is highlighted.
  4. The decimals are corrected and are the uniform throughout the manuscript
  5. ‘EDX’ is deleted
  6. AFM, ambient tapping mode using tip (RTESPW tip (Veeco Manufacturing, Inc.) model Results and discussion
  7. The subtitle in section 3.1 is modified to ‘’Cyclic voltammetry of Au coated glass in Te 4+ and Cu2+ Solutions’’
  8. ’CVs were generated’’ is replaced with ‘’CVs were recorded’’
  9. Subscript and superscripts are fixed in Figure 1
  10. ‘’Initial voltage’’ is replaced with ‘’initial potential’’ throughout in text
  11. The equation 1 and 2 are divided
  12. In line 119 ‘’treated’’ is replaced with ‘’in solution of’’
  13. Line 122, repetition of ‘’interaction’’ is deleted
  14. All abbreviations are defined first, before use
  15. The sentence ‘The results were arrived at using the SEM and AFM techniques’’ in line 117 is removed.
  16. Repetition of word ‘’potentially is deleted.
  17. The root mean square is deleted as suggested, the statically data was calculated on different spots on the sample, (scanning from east to west and from north to south) in the entire sample with more than 20 spots.
  18. In section 3.4 two decimal places were kept as suggested
  19. Forward peak current (if) and backward or reverse peak current (ib) is defined first.
  20. Experimentally XRD pattern is presented in Figure S3
  21. The manuscript was cross-checked by English native personnel. Proofread with an assistance of Grammarly and QuillBot software

Reviewer 2 Report

The manuscript described an alternative electrosynthesis of PdTe by using the electrochemical atomic layer deposition method. The effect of the ratio between Te and Pd on the geometrical and electronic properties of the structure is investigated by combining X-ray diffraction and DFT calculations. Moreover, the binding energies of the oxygen with the different systems in studied to investigate the PdTe catalytic activity. The first part of the manuscript is well presented, while in the final part there are several problems with the numeration of Figures and their description in the text. I suggest the publication after major revisions.

Revisions:

T1) The main problem in this manuscript is the Section 3.6, where all the Figures are cited in the incorrect way in the text (i.e. page 11 line 386 there is not the d-band in the Figure 9 or page 12 lines 415, 417, 427 and so on). It is a very hard work for the reviewer to understand the manuscript with a so large errors on the citation of the figures.

22) At page 11 line 392 authors wrote that “the Pd3Te1 exhibit reduction along x, while elongation along the y”, but the number at line 391 suggests only an elongation of x (4.092 Ang).

33) In the Figure 9, to help the reader, it would be better to include for comparison the experimental results cited at line 395.

44) At page 12, lines 429-430, authors wrote that the results obtained from the d-band center indicate the surface reactivity for ethanol oxidation. A larger explanation is necessary to correlate the d band position and the capability of the surface to oxidate the ethanol.

55)  At page 13, line 474, there is an error with the formula

66)  At pages 13 and 14, authors studied the interaction of the oxygen with the surface at different adsorption site. This choice is quite strange since in the final part of the manuscript the proposed mechanism involved the OH-. The calculation of the OH-binding energies could be a more realistic choice.

77) The sentence “The strong bond formed by oxygen can potentially result in the induction of a catalytic oxidation reaction” is not justified by the presented results. It must be better supported or explained.

88)  In Figure 8 it is not clear the “Relationship between experimentally measured specific activity recorded for ethanol oxi-380 dation and the oxygen binding energies and d-band center values recorded for the cases of Pd (111), Pd3Te2 (111), and Pd3Te (111).”. Which are the experimental values for each case?

99) In the final part of the manuscript, author suggested that the dominant effect is due to the geometrical effect, while at the end od the abstract seems that electronic and geometrical parameters as comparable effects.

Author Response

  1. The figure numeric is fixed and the figure numbers corresponds with in text as highlighted.
  2. T1) The main problem in this manuscript is the Section 3.6, where all the Figures are cited in the incorrect way in the text (i.e. page 11 line 386 there is not the d-band in the Figure 9 or page 12 lines 415, 417, 427 and so on). It is a very hard work for the reviewer to understand the manuscript with a so large errors on the citation of the figures.
  3. The figure number and the legends are fixed throughout the manuscript as heighted.
  4. 22) At page 11 line 392 authors wrote that “the Pd3Te1 exhibit reduction along x, while elongation along the y”, but the number at line 391 suggests only an elongation of x (4.092 Ang).
  5. Answer) Thank you for your careful correction. The typo error in whole manuscript was modified and The sentence was fixed from “The lattice parameters corresponding to Pd3Te1 exhibit reduction along x, while elongation along the y and c-axes can be attributed to the process of replacement of the face site by the Te atoms” to “The lattice parameters corresponding to Pd2Te2 exhibit reduction along x, while elongation along the y and c-axes can be attributed to the process of replacement of the face site by the Te atoms”
  6. 33) In the Figure 9, to help the reader, it would be better to include for comparison the experimental results cited at line 395.
  7. Answer) As accommodate reviewers’ comments, the discussion was added in revised manuscript in “~, expanding lattice parameter with increasing Te contents in the bulk lattice as shown in Figure S3 and S4.”
  8. 44)At page 12, lines 429-430, authors wrote that the results obtained from the d-band center indicate the surface reactivity for ethanol oxidation. A larger explanation is necessary to correlate the d band position and the capability of the surface to oxidate the ethanol.
  9. Answer) I appreciate your kind comment. As demonstrated in Figure. 8, this work obviously shows the correlation of the strength of OH bonding on the catalyst surface and EtOH oxidation. The d-band canter model apparently explained the higher position of d-band weight leads to strong bonding oxygen intermediates because of the increasing unoccupied anti-bonding state after the adsorption of oxygen intermediates. It was described in revised manuscript “The mechanism that can best explain the adsorption and generation of species has been presented. The extent of CH3COads and OHads coverage realized influences the mechanism presented in the supporting information and literature reports. The rate-determining step was identified by analyzing Equation. The OHads units adsorbed on the electrode play an important role during EOR.”
  10. 55)At page 13, line 474, there is an error with the formula
  11. Answer) The error in the equation has been fixed
  12. 66)At pages 13 and 14, authors studied the interaction of the oxygen with the surface at different adsorption site. This choice is quite strange since in the final part of the manuscript the proposed mechanism involved the OH-. The calculation of the OH-binding energies could be a more realistic choice.
  13. Answer) I appreciate the reviews’ kind comments. The DFT calculations go through the process to find a thermodynamic stable structure which calls global minima. Theoretically, there is one structure at specific condition. Therefore, the adsorption sites investigation is very critical to obtain trustable data. Therefore, every possible configuration for each site was investigated thus we can say that our results are reliable.
  14. 77) The sentence “The strong bond formed by oxygen can potentially result in the induction of a catalytic oxidation reaction” is not justified by the presented results. It must be better supported or explained.
  15. Answer) I appreciate your kind criticism. As I answered and discussed above, strong oxygen intermediates on the catalyst surface are due to a higher shift of the d-band canter as shown in Fig. 8. The results were discussed in “The strong bond formed by oxygen can potentially result in the induction of a catalytic oxidation reaction. We observed that the d-band shifted downward, indicating the formation of weak bonds between oxygen species and Pd surface. Low activity is observed for the Pd3Te1 catalyst”
  16. 88) In Figure 8 it is not clear the “Relationship between experimentally measured specific activity recorded for ethanol oxi-380 dation and the oxygen binding energies and d-band center values recorded for the cases of Pd (111), Pd3Te2 (111), and Pd3Te (111).”. Which are the experimental values for each case?
  17. Experimental details are presented in section 3.4 and 3.5, and the caption of this figure is reviewed
  18. 99)In the final part of the manuscript, author suggested that the dominant effect is due to the geometrical effect, while at the end od the abstract seems that electronic and geometrical parameters as comparable effects.
  19. Answer) It is general information to determine the catalytic activity. Our study is well supported that the electronic structural effect is significantly dominant thus the sentence was updated.

Round 2

Reviewer 2 Report

The authors modified the manuscript with the inclusion of all suggestions. The quality of the study is definitively improved, and they dissolved my doubts. Thus, I suggest the publication in the present form.